# Ultrasound on the Frontlines: Empowering Paramedics with Lung Ultrasound for Dyspnea Diagnosis in Adults—A Pilot Study

**DOI:** 10.3390/diagnostics13223412

**Published:** 2023-11-09

**Authors:** Damian Kowalczyk, Miłosz Turkowiak, Wojciech Jerzy Piotrowski, Oskar Rosiak, Adam Jerzy Białas

**Affiliations:** 1Department of Pneumology, 2nd Chair of Internal Medicine, Medical University of Lodz, 90-419 Lodz, Poland; damiankowalczyk@onet.pl (D.K.); wojciech.piotrowski@umed.lodz.pl (W.J.P.); 2Department of Anesthesiology and Intensive Care, National Institute of Medicine of the Ministry of the Interior and Administration, 02-507 Warsaw, Poland; milosztr@gmail.com; 3Department of Otolaryngology, Polish Mother’s Memorial Hospital Research Institute, 93-338 Lodz, Poland; oskar.rosiak@iczmp.edu.pl; 4Department of Pulmonary Rehabilitation, Regional Medical Center for Lung Diseases and Rehabilitation, Blessed Rafal Chylinski Memorial Hospital for Lung Diseases, 91-520 Lodz, Poland

**Keywords:** thoracic ultrasound, point-of-care ultrasound, POCUS, BLUE, eFAST, dyspnea

## Abstract

Lung transthoracic ultrasound (LUS) is an accessible and widely applicable method of rapidly imaging certain pathologies in the thorax. LUS proves to be an optimal tool in respiratory emergency medicine, applicable in various clinical settings. However, despite the rapid development of bedside ultrasonography, or point-of-care (POCUS) ultrasound, there remains a scarcity of knowledge about the use of LUS in pre-hospital settings. Therefore, our aim was to assess the usefulness of LUS as an additional tool in diagnosing dyspnea when performed by experienced paramedics in real-life, pre-hospital settings. Participants were recruited consecutively among patients who called for an emergency due to dyspnea in the Warsaw region of Poland. All the enrolled patients were admitted to the Emergency Department (ED). In the prehospital setting, a paramedic experienced in LUS conducted an ultrasonographic examination of the thorax, including Bedside Lung Ultrasound in Emergency (BLUE) and extended Focused Assessment with Sonography for Trauma (eFAST) protocols. The paramedic’s diagnosis was compared to the ED diagnosis, and if available, to the final diagnosis established on the day of discharge from the hospital. We enrolled 44 patients in the study, comprising 22 (50%) men and (50%) women with a median age of 76 (IQR: 69.75–84.5) years. The LUS performed by paramedic was concordant with the discharge diagnosis in 90.91% of cases, where the final diagnosis was established on the day of discharge from the hospital. In cases where the patient was treated only in the ED, the pre-hospital LUS was concordant with the ED diagnosis in 88.64% of cases. The mean time of the LUS examination was 63.66 s (SD: 19.22). The inter-rater agreement between the pre-hospital diagnosis and ER diagnosis based on pre-hospital LUS and complete ER evaluation was estimated at k = 0.822 (SE: 0.07; 95%CI: 0.68, 0.96), indicating strong agreement, and between the pre-hospital diagnosis based on LUS and final discharge diagnosis, it was estimated at k = 0.934 (SE: 0.03; 95%CI: 0.88, 0.99), indicating almost perfect agreement. In conclusion, paramedic-acquired LUS seems to be a useful tool in the pre-hospital differential diagnosis of dyspnea in adults.

## 1. Introduction

Lung transthoracic ultrasound (LUS) is a relatively young, but widely recognized, method of rapidly imaging certain pathologies in the thorax. Moreover, it is devoid of the risks associated with exposure to ionizing radiation posed by classical methods of chest radiology, making it a safer alternative. Additionally, LUS is more accessible and widely applicable than highly specialized imaging methods. As a result, it proves to be an optimal tool in respiratory emergency medicine, applicable in various clinical settings. However, despite the rapid development of bedside ultrasonography, or point-of-care (POCUS) ultrasound, there remains a scarcity of knowledge about the use of LUS in pre-hospital settings [1]. 

Dyspnea is one of the most common chief complaints in the ED. The differential diagnosis can be challenging, as it includes various disorders, including respiratory and cardiac diseases. Medical imaging is one of the key elements in this diagnostic process. There are three main imaging techniques accessible in the ED: chest radiograph (CXR), computed tomography (CT), and LUS. CXR is often the primary imaging modality; however, it is associated with radiation, as well as seeming to serve a relatively low sensitivity in the diagnosis of acute dyspnea [2]. The low sensitivity of CXR was observed in the diagnosis of pneumothorax, pleural effusion, and pulmonary edema, particularly in bedside-acquired images [3]. By contrast, CT scans offer detailed imaging of the cardiorespiratory system and, therefore, sensitive and specific results. However, it is burdened by a significant radiation exposure and risk of contrast complications [2]. On the other hand, we have LUS, which seems to be a safe, quick, readily available, and accurate tool for the emergency diagnosis of pneumonia, acute heart failure, and exacerbations of COPD or asthma [4,5]. 

Undoubtedly, one of the most important factors increasing a patient’s chances of survival in a life-threatening situation is the time taken to establish a correct initial diagnosis, making decisions regarding initial treatment, and determining the optimal location for its continuation. Therefore, shifting the LUS, with all the advantages it serves, from the ED to pre-hospital settings, is, in fact, moving it to the frontline of the differential diagnosis, and as a result—potentially saving time in emergencies. 

In Poland, the first contact with a patient in an emergency condition will most often be made by a paramedic. Therefore, our aim was to assess the usefulness of LUS as an additional tool in diagnosing dyspnea when performed by an experienced paramedic in real-life, pre-hospital settings.

## 2. Materials and Methods

### 2.1. Participants

The participants were recruited consecutively among patients who called for emergency assistance due to dyspnea in the Warsaw region of Poland and were attended to by the rescue team, which included a paramedic who was a researcher in our study (DK). All the enrolled patients were admitted to the Emergency Department (ED) at Priest Jerzy Popieluszko Memorial Hospital in Warsaw. Informed consent was obtained from all subjects involved in the study. The study protocol was approved by the Ethics Committee of the Medical University of Lodz (protocol No. RNN/69/22/KE). In the pre-hospital setting, one of the authors—a paramedic experienced in LUS (DK)—performed an ultrasonographic examination of the thorax according to the Bedside Lung Ultrasound in Emergency (BLUE) [6] and extended Focused Assessment with Sonography for Trauma (eFAST) [7] protocols. Enrollment took place from 4 April 2022 to 15 June 2023. The only exclusion criterion was the inability to give informed consent to participate in the study. Besides LUS, the researcher performed a structured clinical assessment, including taking a medical history, a full physical examination, and basic emergency diagnostics (electrocardiography, pulse oximetry, capnometry) (see Appendix A). Therefore, our study is a real-life assessment of LUS as an additional tool in the pre-hospital diagnosis of dyspnea. The paramedic diagnosis was compared to the Emergency Department diagnosis and, if available, as a reference to the final diagnosis established on the day of the discharge from the hospital. 

The lungs were imaged using two ultrasonographic probes: a convex probe with a frequency of 2–5 MHz, a field of view of 67.3%, and a scanning plane of 50 mm; and a linear probe with a frequency of 4–12 MHz, a field of view of 34.5 mm, and a scanning plane of 34 mm. The probes were positioned according to the BLUE protocol in at least three locations on each side of the chest, and each time, the images were obtained in both the longitudinal and transverse planes. Based on the obtained images, four main lung imaging profiles were identified in patients with dyspnea: profile A, characterized by the presence of horizontal, symmetric artifacts reflecting the pleural line without other sonographic changes above both lung fields (Figure 1); and profile B, characterized by numerous, merging vertical lines, extending from the pleural line to the end of the image (above three lines in each intercostal space—the Merlin location, the imaging site between two ribs). The classification of B lines could be: B lines > 3, B lines > 7, and confluent B lines. These changes had to be identical above both lungs (Figure 2 and Figure 3). Figure 2 shows an image of confluent B lines; Figure 3 shows an image of B lines > 7; and Profile C features symmetrical bilateral subpleural consolidation changes with uneven shapes, most commonly showing characteristics of atelectasis and/or inflammation (Figure 4). Pleural effusion can be unilateral (e.g., in the context of cancer) or bilateral (e.g., in the context of heart failure), extending from the diaphragmatic line, gravitationally aligning within the pleural cavity (Figure 5 and Figure 6).

### 2.2. Data Collection

Data concerning demography, comorbidities, and clinical characteristics were collected.

The LUS examination included the following anatomical divisions: the lungs, pleura, and pleural lines, which were assessed using either a convex or a linear probe. In accordance with various protocols, POCUS was conducted as follows: the BLUE protocol involved the use of both convex and linear probes, and the eFAST protocol was performed with a convex probe.

For the systematicity of the study and the possibility of comparing the images obtained, all ultrasound examinations were performed with the same ultrasound machine—a Philips Lumify System (Philips, Amsterdam, The Netherlands).

The initial diagnosis made by the paramedic was based on a structured clinical assessment, including medical history, a comprehensive physical examination, and basic emergency vital signs measurements (such as electrocardiography, pulse oximetry, and capnometry), with LUS as an additional tool. Each initial diagnosis was consistently documented in the medical rescue actions card, a part of the patient’s official medical record. This card served as the basis for comparing the paramedic’s diagnosis with the final diagnosis determined in the Emergency Department or on the day of the hospital discharge.

### 2.3. Statistical Analysis

Age was presented as the median and interquartile range (IQR) from LQ (25%) to UQ (75%), due to non-normal distribution, according to the Shapiro–Wilk test. Categorical data were presented as absolute values and percentages. Such data were compared using McNemar’s chi-squared test with continuity correction.

Inter-rater agreement was assessed using Cohen’s kappa coefficient. The diagnosis made by the ER and the pre-hospital diagnosis were compared and Cohen’s kappa was used for the comparison of nominal values. The result was expressed as kappa (κ) with standard error (SE) and a 95% confidence interval (CI). Interpretation followed the recommendations published by McHugh et al. [8].

Analysis was performed using R software v. 4.3.1 (R Core Team (2018). R: A language and environment for statistical computing. R Foundation for Statistical Computing, Vienna, Austria). The Kappa value was calculated using Statisitca 13.1 Software.

## 3. Results

### 3.1. Participants’ Characteristics

We enrolled 44 patients in the study—22 (50%) men and (50%) women, with a median age of 76 (IQR: 69.75–84.5) years. In the LUS, patients presented profile C in 13 (29.55%), profile B in 10 (22.73%), profile A in 6 (13.64%), profile (B/C) in 5 (11.36%), pleural effusion in 4 (9.09%), profile A/B in 3 (6.82%), and profile A/C in 3 (6.82%) cases.

### 3.2. The Effectiveness of LUS in the Prehospital Setting

The LUS performed by the paramedic was concordant with discharge diagnosis in 90.91% of the final diagnoses established on the day of discharge from the hospital. In cases where the patient was treated only in the ED, pre-hospital LUS was concordant with ED diagnosis in 88.64% of cases.

We did not observe a statistical difference between the result of pre-hospital and Emergency Departments in the context of the final diagnosis established on the day of the discharge from the hospital (McNemar’s chi-squared = 0, *p* = 1.0) (Table 1).

The mean time of the LUS examination was 63.66 s (SD: 19.22).

The differences between pre-hospital LUS and the final diagnoses established on the day of the discharge were congestive heart failure diagnosed as pneumonia or vice versa in—3 (9.01%) patients. It is worth mentioning that in one case, such a wrong initial diagnosis was established by the Emergency Department, whereas the pre-hospital diagnosis was correct. This observation confirms a common view on differentiating between these two conditions as being challenging in everyday clinical practice.

Additionally, compared to the Emergency Department assessment only, one pneumonia was overdiagnosed in LUS, and the exacerbation of chronic obstructive pulmonary disease was misdiagnosed with respiratory failure due to ethanol intoxication.

We observed that profiles C and B were the LUS image results that led to misdiagnosis; however, the number of patients is too small to draw any conclusion on this matter (Table 2).

Finally, the inter-rater agreement between the pre-hospital diagnosis and ER diagnosis based on pre-hospital LUS and complete ER evaluation was estimated at κ = 0.822 (SE: 0.07; 95%CI: 0.68–0.96), indicating strong agreement, and between the pre-hospital diagnosis based on LUS and final discharge diagnosis, it was estimated at κ = 0.934 (SE: 0.03; 95%CI: 0.88–0.99), indicating almost perfect agreement.

## 4. Discussion

We aimed to assess the usefulness of LUS when acquired by experienced paramedic for the pre-hospital diagnosis of dyspnea in adults. The primary researcher evaluated the LUS images in a comprehensive clinical context, taking into account safety, ethical considerations, clinical implications, and applicability. Our study provides a practical evaluation of LUS as a diagnostic tool complementary to routine patient assessment for dyspnea in pre-hospital settings. Our findings suggest that this combination is comparable to assessments conducted in the Emergency Department and also highly consistent with the diagnosis made upon discharge from the hospital. Therefore, it highlights the usefulness of LUS in the differential diagnosis of dyspnea in adults. Additionally, the relatively short duration of the examination emphasizes its efficiency as a time-effective tool.

The success rate of lung imaging in the pre-hospital setting in correlation with the acute diagnosis in the hospital setting was 90.91% in our study. This is a very encouraging result, since, on three occasions, ED staff made an incorrect diagnosis of lung disease despite the correct diagnosis made by the paramedic based on LUS.

The current literature on the utility of lung ultrasound conducted by paramedics in pre-hospital settings remains limited. Nevertheless, there is a growing body of evidence supporting this approach. Notably, a study by Brooke et al. demonstrated that ultrasound-naive practitioners can attain a satisfactory level of competence within a brief timeframe through simulation-based training [9]. Also, a noteworthy pilot study conducted in a prehospital setting by Schoeneck et al. revealed good inter-rater agreement in the detection of any B-lines between paramedics and expert oncologists (k = 0.60; 95%CI: 0.36–0.84). The authors of this study concluded that pre-hospital lung ultrasound for B-lines holds promise in aiding the identification or exclusion of congestive heart failure as a potential etiology of dyspnea [10]. It is also noteworthy to mention a cohort study conducted by Scharonow et al., wherein prehospital diagnoses were subsequently confirmed in 90.8% of cases. Importantly, the disparities between pre-hospital and in-hospital findings were not statistically significant [11]. The validity of the use of LUS in the pre-hospital setting is also confirmed by the study by Donovan et al. The authors included a review of publications on POCUS from Ovid MEDLINE, EMBASE, CINAHAL Plus, and PUBMED databases. The authors considered publications from 1 January 1990 to 14 April 2021, including 591 articles, of which only 7 met the inclusion criteria, i.e., related to POCUS diagnosis performed by non-medical personnel. The authors emphasize the common denominator of the articles on POCUS diagnosis, describing the validity of ultrasound examination in the hands of paramedics, the similar results of the examination compared with imaging by medical personnel, and the great potential for applying LUS diagnostics to patients with symptoms of respiratory failure in the pre-hospital setting [12].

Another pertinent perspective in this discourse comes from the systematic review conducted by Swamy et al., which sought to evaluate the ability of nurses, students, and paramedics to accurately identify B-lines and pleural effusions for detecting pulmonary congestion in heart failure, while also examining the necessary training [13]. Regrettably, among the eligible studies, only one investigated paramedic-performed lung ultrasound, and it did not endorse the ability of paramedics to competently acquire and interpret lung ultrasound images after a 2-h training session. This study, seemingly at odds with our findings, is the work of Becker et al. [14]. The authors of this study described the efficacy and validity of the use of LUS by paramedics after a 2-h training course in sub-primary lung ultrasound diagnosis. The diagnosis of lung disease was made in correlation with materials uploaded to the emergency medical services command center (EMS-CC) physician. The conclusions of the study were unfavorable, due to numerous problems in performing the examination, including equipment failures, refusals to perform the examination, and inadequate performance of the LUS diagnosis. Lung ultrasound in the pre-hospital setting, in correlation with the physician’s assessment, was judged unfeasible in the real environment with currently available technologies [14]. We believe that the fundamental problem described in the study by Becker et al. is the inadequate training of paramedics and the fact that there is a lack of individuality in the therapeutic decisions made at the scene requiring the transmission of examination materials—video ultrasound, which greatly hinders communication and the ability to interpret the examination. The need for a longer educational process for paramedics emphasizing LUS is described by Guy et al. [15]. The authors described a two-day ultrasound training system for paramedics covering abdominal, pulmonary, and cardiac diagnostics. All participants passed the practical exam, and the results of the theoretical exam oscillated around a 70% pass rate [15].

In our study, we described relatively large group of patients in whom sonographic signs in LUS were classified on ultrasound as profile C (29.55%), correlating most often with extramural pneumonia. Sonographic signs, i.e., lung consolidations, positive air bronchogram, abnormal pleural line, and pleural effusion as sonographic C-profile, were also described and correlated with X-ray diagnosis during the differential diagnosis of pneumonia in children by Yan et al. [16]. They demonstrated that LUS is a reliable and valuable alternative in the diagnosis of pneumonia. The use of thoracic ultrasound as a first-line diagnostic tool in the diagnosis of patients with suspected pneumonia has also been stressed by Pagano et al. [17], who highlighted the sensitivity of the LUS examination at 0.985 and a specificity of 0.649 in terms of the diagnosis of POCUS pneumonia in the ED based on a study of 107 patients. An additional aspect worth highlighting is the fact that the authors showed a higher sensitivity of LUS examination of the chest for the differential diagnosis of pneumonia compared to the diagnosis of a chest X-ray. Similar findings were reported by Cortellaro et al. [18]. The authors stress the validity of LUS in the differential diagnosis of pneumonia, emphasizing the time efficiency of the examination and the higher sensitivity of the test compared to chest X-rays.

The common denominator of the available publications on LUS is the validation of the use of such diagnostic imaging when treating patients with symptoms of respiratory failure. The majority of authors of papers on LUS stress the validity of such an examination when caring for patients with respiratory failure, emphasizing the high sensitivity of the examination. The authors are unanimous in emphasizing the validity of standardized and consistent training processes for LUS to obtain the best possible results from the test. We also agree with such conclusions in our publication.

Due to the small number of studies describing the validity of using LUS in pre-hospital diagnostics during the differential diagnosis of dyspnea, a valuable voice in the discussion is the publication by Costantino Caroselli and Antonio Cherubini. The authors describe the feasibility of using LUS in pre-hospital conditions, e.g., in a nursing home during the sonographic diagnosis of patients with COVID-19 disease. The authors emphasize that LUS diagnostics can reduce the percentage of unnecessary hospitalizations and play a significant role in predicting mortality in a group of elderly patients with COVID-19 [19].

LUS offers several advantages, including safety, time efficiency, ready availability, and accuracy, making it an effective tool for diagnosing pneumonia, acute heart failure, and exacerbations of COPD or asthma in emergencies [4,5]. It also has the potential to expedite on-site patient assessments and facilitate targeted pre-hospital treatment, followed by the seamless continuation of care in the ED.

A valuable contribution to the discussion is also the study by Rocca et al. The authors showed that LUS influences clinical decision-making by changing the diagnostic process and patient management. However, they highlight concerns about the skill of POCUS diagnosis and the rationale for setting a pathway for adequate education [20]. Değirmenci et al. investigated the usefulness of a short course conducted on ultrasound simulators in teaching the Focused Assessment with Sonography for Trauma (FAST) protocol. The authors enrolled 60 participants in the study, including 20 paramedics, who achieved 98% image correctness and 81.5% correct diagnoses [21]. To the best of our knowledge, this report is the only one in the literature that includes a cohort of paramedics. It underscores the importance of not only providing practical skills development, but also a robust foundation of theoretical knowledge to ensure high diagnostic accuracy. Consequently, considering both the advantages and didactic challenges of teaching Lung Ultrasound (LUS), we propose the implementation of an intensive, mandatory LUS training program for paramedics. This training should encompass both theoretical and practical aspects, with an emphasis on an in-depth knowledge of clinical differential diagnosis principles. Ideally, such training should commence in the pregraduate phase and continue in postgraduate education, accompanied by a well-structured certification system. Considering all the points mentioned above, this approach appears beneficial from both clinical and healthcare policy perspectives.

The primary constraint in our study is the relatively small number of participants, which characterizes our study as a pilot study and should be considered hypothesis-generating. This limitation arises from the involvement of a single skilled paramedic ultrasound examiner responsible for conducting examinations; on the other hand, it enhances the reproducibility and standardization of the LUS examinations. Furthermore, the patients included in our study often presented life-threatening conditions, which added to the challenge of recruiting a larger sample. Additionally, we excluded unconscious patients from our study, further narrowing our target group. Collectively, these factors posed difficulties in assembling a sizable study cohort. Another noteworthy concern pertains to the experience of the paramedic who performed LUS in our study. He possesses extensive expertise in LUS, having conducted pre-hospital diagnostics for 36 months and holding certifications to validate his qualifications. Additionally, the integration of LUS into official medical records and the communication of its results to the emergency physician in the ED could introduce potential biases. Nevertheless, it is worth noting that conducting such a study in a randomized, blinded manner for ethical and legal reasons appears to be unfeasible.

## 5. Conclusions

In conclusion, paramedic-acquired LUS seems to be a useful tool in the pre-hospital differential diagnosis of dyspnea in adults.

## Figures and Tables

**Figure 1 diagnostics-13-03412-f001:**
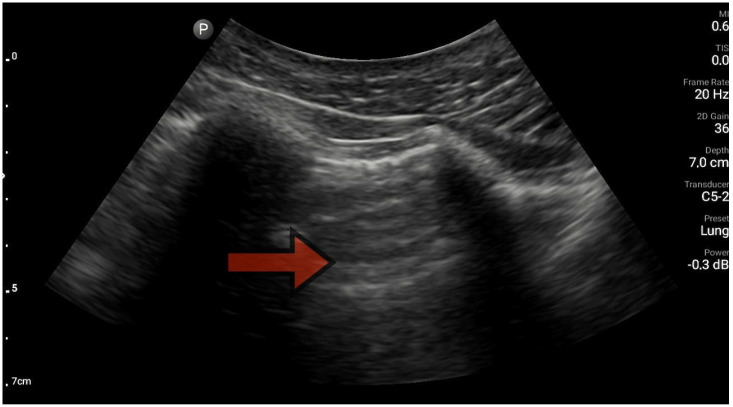
Profile A, line A (source: author’s material—DK).

**Figure 2 diagnostics-13-03412-f002:**
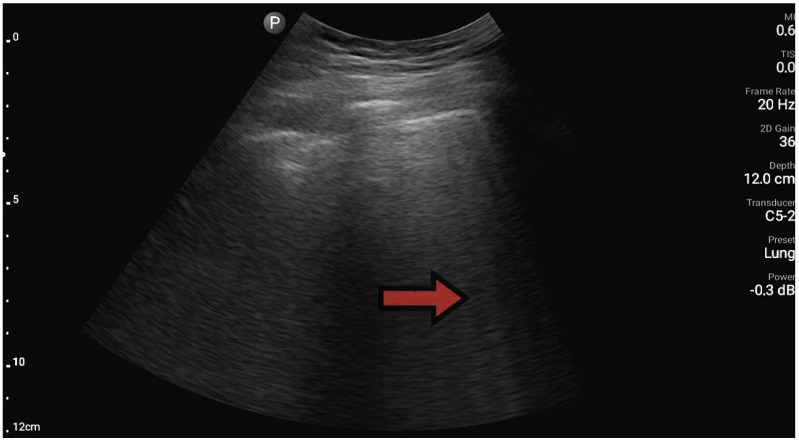
Profile B, line B (source: author’s material—DK).

**Figure 3 diagnostics-13-03412-f003:**
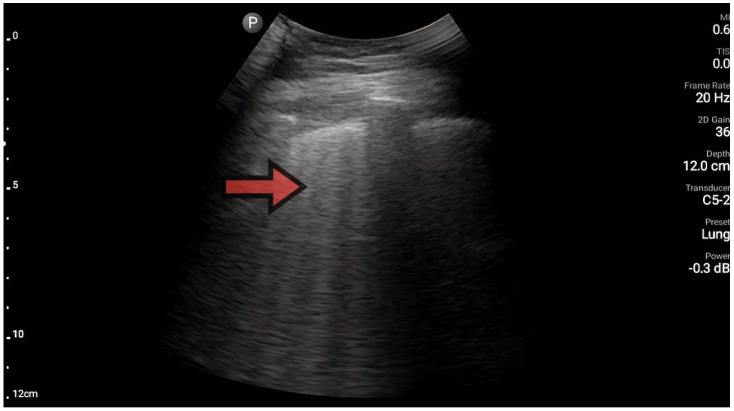
Profile B, line B (source: author’s material—DK).

**Figure 4 diagnostics-13-03412-f004:**
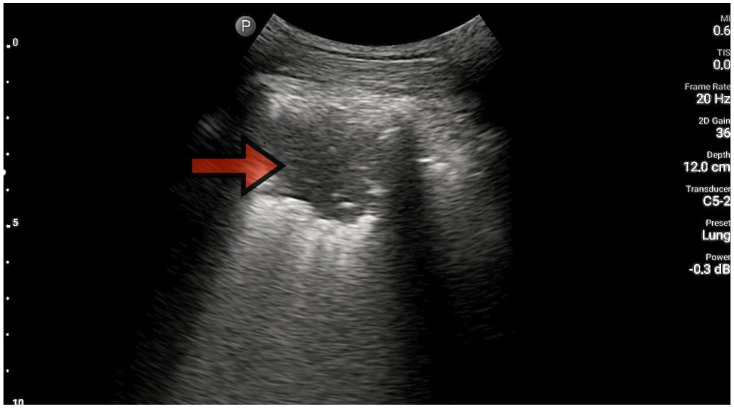
Consolidation (source: author’s material—DK).

**Figure 5 diagnostics-13-03412-f005:**
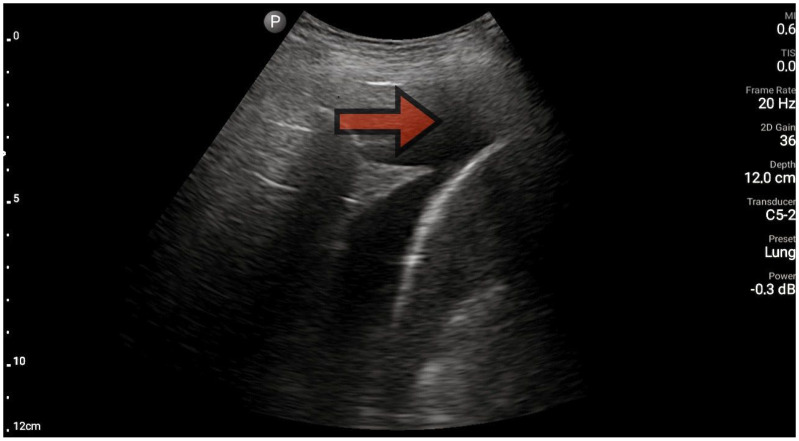
Pleural effusion (source: author’s material—DK).

**Figure 6 diagnostics-13-03412-f006:**
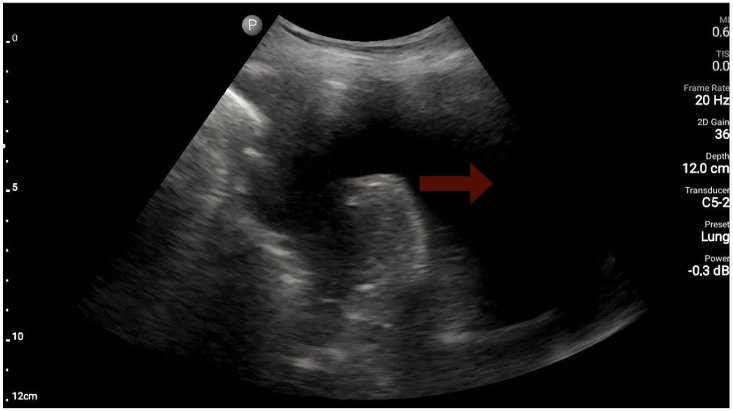
Pleural effusion (source: author’s material—DK).

**Table 1 diagnostics-13-03412-t001:** The comparison between pre-hospital LUS and ER assessment in terms of their concordance with the final diagnosis was established on the day of the discharge from the hospital. The *p*-value is for McNemar’s chi-squared test with continuity correction.

	ER Concordant with Final Diagnosis	ER Not Concordant with Final Diagnosis	*p*-Value
Pre-hospital LUS concordant with final diagnosis	27	2	1.0
Pre-hospital LUS not concordant with final diagnosis	3	0	

**Table 2 diagnostics-13-03412-t002:** The basic characteristics of the pre-hospital LUS and their relations with the concordance with the final diagnosis are established on the day of the discharge from the hospital.

Observation	Diagnosis Concordant *n* = 30 (90.91%)	Diagnosis Unconcordant *n* = 3 (9.09%)
Profile A	2 (6.67%)	0 (0%)
Profile A/B	2 (6.67%)	0 (0%)
Profile A/C	2 (6.67%)	0 (0%)
Profile B	7 (23.33%)	1 (0.33)
Profile B/C	4 (13.33%)	0 (0%)
Profile C	9 (30%)	2 (0.67)
Pleural effusion	4 (13.33%)	0 (0%)

## Data Availability

The data used and/or analyzed in this study are available from the corresponding author upon reasonable request.

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
