# Peer review of "Ultrasound on the Frontlines: Empowering Paramedics with Lung Ultrasound for Dyspnea Diagnosis in Adults—A Pilot Study"

_diagnostics, 2023, doi:10.3390/diagnostics13223412_

Round 1
Reviewer 1 Report
Comments and Suggestions for Authors
Thank you for the opportunity to review your manuscript. I find your research valuable in the field of pre-hospital emergency medicine, and innovative, as it focuses on the use of LUS by paramedics to diagnose possible causes of dyspnoea in adult patients. Your study's findings suggest the potential of LUS performed by paramedics to provide accurate diagnoses and to improve the efficiency and accuracy of emergency care. Your research also suggests that LUS can be performed quickly without compromising diagnostic accuracy. Overall, your manuscript is clear, concise, and well-structured, making it easy for readers to understand your pilot study's objectives, methods, and key findings.
Author Response
Dear Reviewer
Thank you very much for your motivating review and for taking the time to read the manuscript.
Based on the remaining responses, we have modified the manuscript and changed the images corresponding to the sonographic pathologies in LUS. We encourage you to familiarize yourself with the changes, we believe that the work is now clearer and the projections are clearer and clearer for the reader.
We have also provided a detailed description of what B lines are and how to recognize them during the examination.
Regards
Damian Kowalczyk
-------------------------------------------
Reviewer 2 Report
Comments and Suggestions for Authors
This study is potentially interesting but I my opinion the authors need to revise some parts of it.
I have some doubts: which experience did the paramedics, who carried out the lung ultrasound, have? Did they attend a certified course? How many expertly supervised tests have they done? Was a lung ultrasound performed on the same patients in the emergency department?
A limit of this study is the low number of patients enrolled. It would be interesting to have a broader case study.
The important role of lung ultrasound in emergency/urgency (for example during the recent SARS CoV2 pandemic) is well defined and it could be useful to include some other recent scientific paper in the bibliography (e.g. C. Caroselli et al, J Ultrasound Med 2022;41:2547–255; Norbedo S et al. J Ultrasound Med. 2020;10).
Why was the image of the pleural effusion obtained with a linear probe? Figure 4 is cropped and the extension of the pleural effusion is not entirely visible: I recommend to use a convex probe to visualize the pleural effusion better. The quality of the other figures is poor and unclear, I recommend to attach to the paper better quality figures among those obtained by the paramedics.
Furthermore I recommend to describe better the artifacts shown in the figures (e.g. in figure 1 the horizontal artefacts have long been defined as A lines, the term comet tails has been abandoned some time ago. How many B lines described in figure B are there for each lung space? In literature exist a classification of B lines > 3, B lines > 7 or confluent B lines: it is important to specify this. In figure C can you show better (for example with arrows) where the consolidation areas are?
Comments on the Quality of English Language
I recommend a revision of the English language and the correction of spelling errors (example: “encouragnig” line 180 on page 6)
Author Response
Dear Reviewer!
Thank you very much for your review and tips on editing this manuscript. I agree with many aspects of your comments, we have made changes to the manuscript regarding sonographic projections (new photos) and descriptions of ultrasound images, including pathology descriptions. I think the manuscript benefited a lot from this change. Unfortunately, the quality of the photos is a big problem, I am aware of that. This is due to the examination with a mobile ultrasound and the acquisition of photos from an existing ultrasound recording (video).
The availability of research on the use of ultrasound in pre-hospital conditions is low, so we would like to thank you for recommending the study and we have supplemented the manuscript with additional citations.
I am also sending answers to your questions below:
The paramedics participating in the study (as described in the manuscript) are trained ultrasound researchers after completing certified courses in abdominal ultrasound, lung ultrasound and ECHO. These trainings were conducted and supervised during patient examinations by specialists in surgery, pulmonology and cardiology, respectively, with ultrasound certification e.g. WINFOCUS Course. Additionally, the paramedics participating in the study have advanced medical qualifications, such as instructor qualifications in the field of ALS.
As researchers working in prehospital medicine, we could not influence decisions regarding imaging tests performed in the ED, so each patient was examined and ultrasounded prehospital, which did not always occur later in the ED. Many doctors and paramedics do not use ultrasound diagnostics and chose X-ray or computed tomography diagnostics in the cases of our patients. We later compared the results of these studies with our ultrasound examinations and described the conclusions drawn from them, e.g. correlations of changes visualized in ultrasound with the description of computed tomography.
With this study, we also want to show the role of ultrasound in pre-hospital conditions and convince undecided people to use this diagnostic method, especially when diagnosing patients in a serious clinical condition associated with shortness of breath. We agree that the research group is not large, this is due to many factors. Firstly, we conducted the research in one center in cooperation with one hospital. In the study, we describe patients who were often in a very serious condition, many of them were unable to sign consent to the study due to impaired consciousness or unconsciousness and had to be excluded from participating in the study. For this reason, we had to exclude a large number of patients. In the future, we plan to organize a similar examination in several centers at the same time, but this requires prior training of advanced paramedics in ultrasound, and unfortunately there are few such people in Poland.
Additionally, we conducted a thorough review of the English language.
Regards!
Damian Kowalczyk
=========================================
Round 2
Reviewer 2 Report
Comments and Suggestions for Authors I think that the corrections made to the manuscript may be sufficient.
I recommend to modify the arrows contained in the figures, replacing them with
small red arrows so as not to cover the figures
Author Response
Thank you for your message! As recommended, I changed the size and color of the arrows in the photos.
Regards!